# Effect of *Limosilactobacillus reuteri* LRE02–*Lacticaseibacillus rhamnosus* LR04 Combination on Antibiotic-Associated Diarrhea in a Pediatric Population: A National Survey

**DOI:** 10.3390/jcm9103080

**Published:** 2020-09-24

**Authors:** Lorenzo Drago, Gabriele Meroni, Antonio Chiaretti, Nicola Laforgia, Salvatore Cucchiara, Maria Elisabetta Baldassarre

**Affiliations:** 1Laboratory of Clinical Microbiology and Microbial Metagenomic Unit, Department of Biomedical Sciences, University of Milan, 20133 Milan, Italy; gabriele.meroni@unimi.it; 2Pediatric Research Center Romeo and Enrica Invernizzi, University of Milan, 20122 Milan, Italy; 3Pediatric Emergency Department, Fondazione Policlinico Universitario A. Gemelli, IRCCS–Rome, 00168 Rome, Italy; achiaretti@yahoo.it; 4Department of Biomedical Science and Human Oncology-Section of Neonatology and NICU, University “Aldo Moro” of Bari, 70124 Bari, Italy; nicola.laforgia@uniba.it (N.L.); mariaelisabetta.baldassarre@uniba.it (M.E.B.); 5Pediatric Gastroenterology and Liver Unit, Department of Women’s and Children’s Health, Sapienza University of Rome, 00161 Rome, Italy; salvatore.cucchiara@uniroma1.it

**Keywords:** antibiotic-associated diarrhea, *Lacticaseibacillus rhamnosus*, *Limosilactobacillus reuteri*, probiotics, antibiotics, children

## Abstract

Probiotics are living microorganisms, which, upon oral ingestion, may prevent antibiotic-associated diarrhea (AAD) through the normalization of an unbalanced gastrointestinal flora. The objective of this study was to evaluate the benefits of a probiotic combination (*Limosilactibacillus reuteri* LRE02-DSM 23878 and *Lacticaseibacillus rhamnosus* LR04-DSM 16605) on the prevention of AAD in an outpatient pediatric setting. Questionnaires were delivered to pediatricians by each patient/parent during the visits after antibiotics and probiotics treatment to monitor physiological parameters. The primary outcome of both groups (probiotics and no probiotics treated) was the evaluation of the prevalence of AAD between the two groups. Evaluation of stool consistency using the Bristol Stool Scale (BSS) score was performed, as well as the evaluation of AAD duration, frequencies of daily evacuation, and the beginning of diarrhea and weight loss during AAD in both groups and related to antibiotic categories. Results indicated that probiotics, at the recommended dosage of 1.2 × 10^9^ CFU (Colony Forming Unit) per day for 30 days, are associated with lower rates of AAD and a decreased number of days with diarrhea, independent of the type of antibiotic used. Moreover, the use of probiotics resulted in a normal stool consistency in a shorter time period, as evaluated by the BSS.

## 1. Introduction

Antibiotic-associated diarrhea (AAD) is defined by the presence of diarrhea with three or more soft/liquid stool evacuations within 24 h or an acute and evident change of fecal pattern during antibiotic treatment or also a few weeks later, without other demonstrable causes [1].

Different mechanisms may determine AAD: a direct toxic effect of antibiotics on the intestine, an altered digestive function secondary to reduced concentrations of gut bacteria, and the over-growth of pathogenic microorganisms [2]. Mostly, it is a mild condition requiring no treatment, and resolving on its own. More-serious AAD instead needs more appropriate monitoring and specific medications [2,3]. An extreme consequence of AAD is the overgrowth of potentially pathogenic organisms, such as *Clostridium difficile* [3].

AAD occurs in about 5–30% of patients either early during antibiotic therapy or up to two months after the end of the treatment [4,5,6].

The frequency of antibiotic-associated diarrhea depends on the antibiotic type (higher risk for penicillins, especially in combination with clavulanate, cephalosporins, and clindamycin), dosage, number of antibiotic prescriptions as well as by host factors (age, prematurity, hospitalization, season) [7].

In the pediatric population, AAD represents a healthy concern being that the burden and costs are also not well documented by national surveillance studies [8]. The AAD in patients treated with antibiotics occurred with an average of up to 30%, with a range of 11% to 62% [9].

Recent trials and metanalysis suggest that specific probiotics are useful in AAD prevention in children [10,11]. 

Goldenberg et al. (2015) performed a systematic literature review, which reported moderate evidence for a protective effect of probiotics in preventing AAD, with an NNT (number needed to treat) of 10. Among the various probiotics evaluated, *Lacticaseibacillus rhamnosus* (formerly *Lactobacillus rhamnosus*) or *Saccharomyces boulardii* at 5–40 billion colony forming units (CFUs)/day appeared to improve NNT [11]. 

In a recent review, Guo et al. (2019) added valuable information on AAD prevention by probiotics. In particular, dosage appeared to account for the substantial reduction in the NTT. The authors also reported the subgroup effect based on high dose probiotics (≥5 billion CFUs per day), which revealed an NNT of six [12]. Overall, these results confirm the importance of strain specificity and dose-related effect of probiotics, as shown by McFarland et al. (2018) [13].

The present work aims to evaluate, in a primary care setting, the effect of a combination of two microencapsulated probiotics on AAD in a vast cohort of children who underwent antibiotic therapy. The probiotic mixture (patented by Probiotical SpA, Novara, Italy) of *Limosilactobacillus reuteri* LRE02–*Lacticaseibacillus rhamnosus* LR04 (1.2 × 10^9^ CFU of both microorganisms daily, 5 drops per day for 30 days) (Abiflor Baby, Aurora Biofarma R&D) was chosen respecting the new rules recently established by a new European Society of Gastroenterology, Hepatology, and Pediatric Nutrition (ESPGHAN) position paper [14], i.e., do not contain plasmids carrying antibiotic resistance; bacteria are gastroprotected thanks to the microencapsulation technology; shelf life (i.e., live and viable bacteria on the expiry date of the product) guaranteed for 24 months; the product is “allergen-free”; it does not contain sucrose or fructose.

## 2. Materials and Methods

### 2.1. Study Design

A survey methodology was conducted for collecting data reported by all the Surveyflor Group, which includes 68 general pediatricians located at the national territory (Italy). The study had been previously approved by the Institutional Review Board, in which many pediatricians in the national territory were included. The survey consisted of a specific questionnaire administered to each patient/parent and delivered to the medical doctor (MD) who collected the data in an electronic online repository. Based on the anamnestic, clinical status, and the outcome of the survey described below, MDs recruited and collected information from the parents and during the visits of each patient enrolled. Inclusion criteria are the following: age 3 years or less, antibiotic therapy less than 24 h ago, written informed consent. Exclusion criteria: acute and/or chronic diarrhea before enrollment, diagnosis of chronic gastrointestinal diseases or diagnosis of other chronic diseases, immunodeficiencies, use of probiotics in the previous two weeks before enrollment, administration of antibiotics 4 weeks before enrollment, prematurity.

### 2.2. Questionnaire and Outcomes

Longitudinal study design, administered online, included patients who underwent antibiotic therapies for different infections (see Table 1). All patients included in the study were randomized by the electronic system to receive antibiotics plus the simultaneous administration of probiotic combination *Limosilactobacillus reuteri* LRE02 (DSM 23878, 2 × 10^8^ CFU daily) and *Lacticaseibacillus rhamnosus* LR04 (DSM 16605, 1 × 10^9^ CFU daily) (Antibiotic–Probiotic group: A), or only antibiotics (Antibiotic group: B). The probiotic combination manufacturer recommended a dosage of five drops/day (equivalent to 1.2 × 10^9^ CFU daily) for 30 days. The AAD episodes in group A and group B were recorded for one month. The questionnaires were delivered by each patient/parent during the visits after 5 (Time 0: T0) and 30 days (Time 1: T1) from the end of antibiotic administration. The primary outcome was the evaluation of the prevalence of AAD between the two groups. Secondary outcomes were: (1) evaluation of stool consistency using the Bristol Stool Scale (BSS) score [15], in both groups in relation to antibiotic categories, and at the end of antibiotic administration (T0) and thirty days after (T1); (2) evaluation of AAD duration, frequencies of daily evacuation, beginning of diarrhea and weight loss during AAD. The Bristol Stool Scale is a diagnostic medical tool designed to classify the form of human feces into seven categories [16]. It is widely used as a research tool to evaluate the effectiveness of treatments for various diseases of the bowel, as well as a clinical communication aid, including being part of the diagnostic triad for irritable bowel syndrome. The seven types of stool are:Type 1: Separate hard lumps, like nuts (difficult to pass and can be black);Type 2: Sausage-shaped, but lumpy;Type 3: Like a sausage but with cracks on its surface (can be black);Type 4: Like a sausage or snake, smooth and soft (average stool);Type 5: Soft blobs with clear cut edges;Type 6: Fluffy pieces with ragged edges, a mushy stool (diarrhea);Type 7: Watery, no solid pieces, entirely liquid (diarrhea).

Types 1 and 2 indicate constipation, with 3 and 4 being the ideal stools as they are easy to defecate while not containing excess liquid, 5 indicating lack of dietary fiber, and 6 and 7 indicating diarrhea.

### 2.3. Statistical Analysis

The statistical analyses were performed with GraphPad Prism version 8.01 for Windows (GraphPad Software^®^, San Diego, CA, USA). The following tests were performed to check the normality (or non-normality) of the distributions: D’Agostino and Pearson omnibus normality test, Shapiro–Wilk normality test, and Kolmogorov–Smirnov normality test. Following these tests, one-way ANOVA and multiple comparisons (Tukey’s multiple comparisons test) were done to compare groups. For non-parametric distributions, the Mann–Whitney test was used. Otherwise, the standard Student’s *t*-test was performed. The Chi-squared test was used to compare the prevalence between the groups. A *p*-value less than 0.05 was considered statistically significant.

## 3. Results

### 3.1. Overall Data

The study included 9960 patients ranging between 1 month and 18 years old, with a mean age (±SD) of 28 ± 21 months. 

Group A (antibiotics plus probiotics) included 5048 patients (50.7% of the total), while group B (antibiotics only) included 4912 patients (49.3%). Infections requiring antibiotic prescription are reported in Table 1 and Table 2, and the main antibiotics prescribed are penicillins, cephalosporins, or macrolides.

Table 2 shows the prevalence of each clinical condition between the two groups. Statistical significances were observed in upper respiratory tract infections (URTI) (*p* value: 0.0357), otitis (Ot) (*p* value: 0.0238), and gastroenteritis (Ge) (*p* value: 0.0002).

Stool consistency results are reported in Figure 1. At T0, in group A the mean value of the BSS score was 4.5 ± 1.5 (median 4, IQR 3–5), while in group B was 4.9 ± 1.6 (median 5, IQR 4–6) (*p* value < 0.0001). Thirty days after (T1), the mean value in group A was 3.7 ± 1.2 (median 4, IQR 3–4), and in group B was 4.2 ± 1.4 (median 4, IQR 3–5) (*p* < 0.0001).

In group A, diarrhea occurred in 1.7 days following treatment, while in group B this occurred after 2.1 days (*p* < 0.0001, Mann–Whitney test). Besides, the number of days with diarrhea in group A (2.8 ± 1.3 days) was significantly lower than those reported in group B (3.2 ± 1.4 days) (*p* < 0.001) (Figure 2A). The number of stools per day was similar between the two groups (3.5 ± 1.8 for group A, and 3.5 ± 1.2 for group B) (Figure 2B). The direct consequence of diarrhea was weight loss after every single episode (Figure 2C). Group A showed a mean weight loss of 320 ± 308 g and group B of 303 ± 235 g, without any statistical difference between the two groups.

Among all the three physiological parameters reported in Figure 2, a stratification between infants, children, and adolescents was made to study the weight loss between the two groups (Appendix A), resulting in no differences between the treated and the control group.

In AAD patients (N = 4897), stool consistency was monitored all throughout the antibiotic administration and also at T0 and 30 days after (T1). The results, summarized in Table 3 and Table 4, show statistical differences between the two groups at T1, with a better improvement of the mean BSS score in group A than in group B. 

### 3.2. Data Analysis after Antibiotic Stratification

The prevalence of prescribed antibiotics among group A and group B is summarized in Table 5. The antibiotics belong to three main pharmacological classes. Only the prevalence of macrolides resulted in statistical differences between the two groups.

For all the three pharmacological categories of antibiotics, the prevalence of diarrhea was statistically lower in group A than in group B (*p* < 0.001) (Figure 3). For more details, see Appendix A.

Regarding stool frequency, the BSS showed at T0 and T1, for all the three main antibiotic categories, a higher score in group B than in group A (Figure 4) except for macrolides (no statistical differences for group B).

The duration (in days) of diarrhea (Figure 5A) was higher in group B for all the three antibiotic categories (*p* < 0.001 for penicillins and cephalosporins, *p* = 0.017 for macrolides); macrolides showed the maximum average of duration (3.3 ± 1.3, mean ± SD) compared to penicillins (3.1 ± 1.4) and cephalosporins (3.2 ± 1.4). The number of stools/day (Figure 5B) was similar between the groups and no differences were found between penicillins and cephalosporins, while macrolides showed difference (*p* = 0.035). Diarrhea in group A started earlier compared to group B regarding penicillins (*p* = 0.01) and cephalosporins (*p* = 0.013), while no differences were found for the macrolides (na = not applicable) (Figure 5C).

The stool consistency in AAD-positive patients (N = 4897) at T0 and T1 shows statistical differences between the two groups with respect to the three antibiotic categories (Table 6). Comparison between two groups was made using a Student’s *t*-test.

## 4. Discussion

The World Health Organization defines probiotics as “live microorganisms which when administered in adequate amounts confer a health benefit on the host” (Food and Agriculture Organization and World Health Organization, 2001). Dr. Metchnikoff opened the “era” of probiotics because he was the first to propose that ingesting certain bacteria could help replace harmful microbes in the body [17].

As reported in the literature, the minimum quantity sufficient to obtain temporary colonization of intestines by a microbial strain is 10^9^ live cells per day [18]. Therefore, in the case of miscellanea, the recommended daily consumption should contain 10^9^ live cells of at least one of the strains, including *Saccharomyces cerevisiae*, or lactic acid fermenters, such as *Streptococcus thermophilus* or *Lactobacillus bulgaricus* [18,19].

Lactobacilli, recently reclassified in 23 different genera [20] and bifidobacteria, are the most common microorganisms used in probiotic preparations. Furthermore, the various products differ in the production processes, which can influence some fundamental characteristics of the probiotic, such as the concentration of viable microorganisms or the presence of contaminants [19,21]. The effects of probiotics are strain- and dose-dependent [1,13]. 

The use of probiotics is now widespread both for intestinal and extraintestinal disorders, especially in pediatrics [3,22,23]. A recent statement of the Working Group on Probiotics and Prebiotics of the European Society of Gastroenterology, Hepatology, and Pediatric Nutrition (ESPGHAN) [1] recommends the use of some probiotic strains for the prevention of AAD. This condition, while transient, is the cause of further morbidity, complications, and sometimes hospitalization with its repercussions in the scope of lost workdays, healthcare costs, and especially baby health [8].

Children are estimated to use three times more antibiotics than adults [24] and AAD seems to occur roughly in 25% of children between the initiation of antibiotics and two months after their completion [25]. Moreover, the use of antibiotics in the first years of life interferes with the microbiota development and results in dysbiosis [26]. A delicate balance exists between the intestinal microbiota and the host; indeed, all conditions causing an imbalance can potentially result in the occurrence of gastrointestinal or extraintestinal diseases [27].

It is also noteworthy that dysbiosis related to antibiotic use in the first years of life is a risk factor for obesity [28], functional gastrointestinal disorders [29] and impaired neurocognitive outcome [30]. 

A Probiotics’ mechanism of action is exerted through the modulation of the content of gut microbiota, maintenance of the integrity of the gut barrier, prevention of bacterial translocation and the modulation of the local immune response by the gut-associated immune system. In particular, *Lacticaseibacillus rhamnosus* is a probiotic strain, which when compared to other probiotics, is one of the most appropriate for preventing AAD due to its modest NNT [31]. *Limosilactobacillus reuteri* has even been extensively studied in several intestinal conditions, and its therapeutic and preventive effects have been documented (e.g., protection from pathogen colonization, decrease in negative interaction due to environmental stressors) [31].

In our randomized intervention study, we evaluated the efficacy of a new double-strain probiotic formulation which included *Limosilactobacillus reuteri* (formerly *Lactobacillus reuteri* LRE02) 2 × 10^8^ CFU and *Lacticaseibacillus rhamnosus* (formerly *Lactobacillus rhamnosus* LR04) 1 × 10^9^ CFU on the prevalence of AAD. These probiotic strains are free of plasmids that carry antibiotic resistance, are microencapsulated, and have a shelf life (i.e., live and viable bacteria on the product expiration date) that is guaranteed for 24 months; moreover, in vitro studies have demonstrated a synergistic ability of these two strains when simultaneously cultivated. The product is “allergen-free” [32] and without sucrose or fructose. Microencapsulation increases the resistance of probiotic microorganisms during the gastro-duodenal transit, thus ensuring their titer and biological activity [33,34]. At the same time, it protects cells from degradation phenomena due to external factors (humidity, acidity, osmotic pressure, oxygen, and light) [33]. One of the advantages using the microencapsulated probiotics is the ability to colonize the intestine in a greater concentration (five times more) compared to non-encapsulated probiotics [35]. As stated in two different studies by Del Piano et al. (2011 and 2012), the microencapsulation of probiotics offers numerous advantages compared to the freeze-dried technique; the most important of which is the high gut colonization as a result of the increased number of viable cells that transit the intestine and, as a consequence, the reduction in the probiotic concentration delivered in these lipidic microcapsules due to a strong gastro-resistance of the cells [33,34].

In our study, the prevalence of AAD was about 50%, regardless of the antibiotic used. This high prevalence, however, previously reported [7], could be related to the low age of included patients (mean age 28 months). In fact, one of the risk factors for AAD is age < 6 years [36].

The most prescribed antibiotics, to treat a range of infections reported in Table 1, were penicillins, followed by cephalosporins and macrolides, according to the recent AIFA (Agenzia italiana del farmaco) National report [37].

Our results showed a significant reduction in AAD prevalence in group A (probiotic plus antibiotics) (38.5%), compared to the control group B (antibiotics alone) (59.9%).

The type of antibiotics and the reason for their administration do not appear to affect the outcome of the probiotic on AAD, as previously reported for one of two probiotics recommended by ESPGHAN for AAD prevention [1,38].

Stool consistency, evaluated with BSS score, was significantly better in group A (mean BSS score 4.5 ± 1.5) than in group B (mean BSS score 4.9 ± 1.6), for all antibiotic categories, both at the end of antibiotic therapy (T0) and after 30 days (T1). In AAD patients, the mean BSS score at T0 was not different between the two groups, regardless of the antibiotic categories used; however, at T1, the BSS score was significantly better in group A than in group B. Surprisingly, in AAD patients, diarrhea started earlier in group A than in group B, regardless of the type of antibiotic used. This effect could be related to the influence of gastrointestinal commensals on motility as previously observed in vivo and ex vivo in mice and rat models, in which the administration of *L. reuteri* and different *Lactobacillus* species were shown to moderate jejunal motility within minutes [39,40,41]. Another explanation could be the interaction between probiotics and smooth muscle of the intestine as stated by Guarino et al. (2008), which studied an in vitro model of human colonic cells exposed to *L. rhamnosus* GG and found a significant shortening of smooth muscle cells that had an impact on motility [42]. Evidence is accumulating on the hypothesis that certain probiotic strains have acute actions in vivo and ex vivo on the host’s autonomic reflexes and can act differently on the small compared to the large intestine [40]. These effects can occur within minutes, suggesting that the inter-kingdom signaling responsible for them do not rely on colonization, alteration in the microbiome composition, or any other longer-term adjustments [43].

However, the number of days with diarrhea was significantly lower in group A than in group B (2.8 ± 1.3 days vs. 3.2 ± 1.4 days), depending on the antibiotic used, as macrolides showed the worst duration. We did not find any difference in the number of stools/day and weight loss in AAD patients in either of the two groups. This is in line with the other studies that have shown that probiotics are associated with a reduction in the mean duration of AAD of 18 h, without having an effect on the number of stools per day [10,23].

Our study is not a placebo-controlled study, and a limitation could be the self-reported symptoms made by parents. However, the large number of patients enrolled, and the utilization of the BSS certainly helps to reduce possible biases.

This study has demonstrated that using our probiotic combination from the beginning of antibiotic therapy is also helpful in reducing the mean duration of diarrhea (calculated in days) as well as in obtaining a normal stool consistency in a shorter time.

## 5. Conclusions

The probiotic mixture of *Limosilactobacillus reuteri* LRE02–*Lacticaseibacillus rhamnosus* LR04, at the recommended dosage of 1.2 × 10^9^ CFU of both microorganisms daily for 30 days, is associated with lower rates of AAD in children (aged one month to 18 years). A reduction in AAD prevalence, the reduced number of days with diarrhea and a better stool consistency were demonstrated regardless of the type of antibiotic used or the reason for their prescription. However, a double-blinded clinical trial is required to further confirm these results.

## Figures and Tables

**Figure 1 jcm-09-03080-f001:**
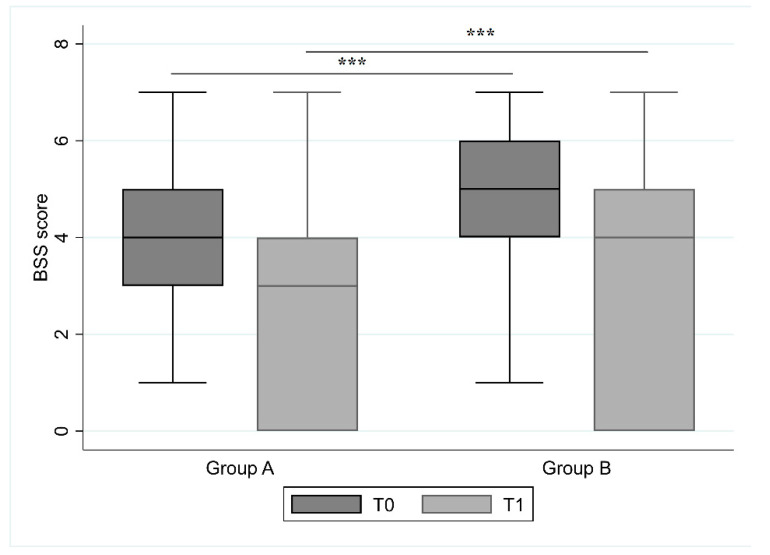
Bristol Stool Scale (BSS) score for evaluation of stool consistency in group A and B at the end (T0) and 30 days after antibiotic treatment (T1). The comparison between two groups at T0 and T1 was made using a Student’s *t*-test and underlined a string statistical difference (*p* < 0.0001) in each of the two groups. (*** *p* value: <0.001).

**Figure 2 jcm-09-03080-f002:**
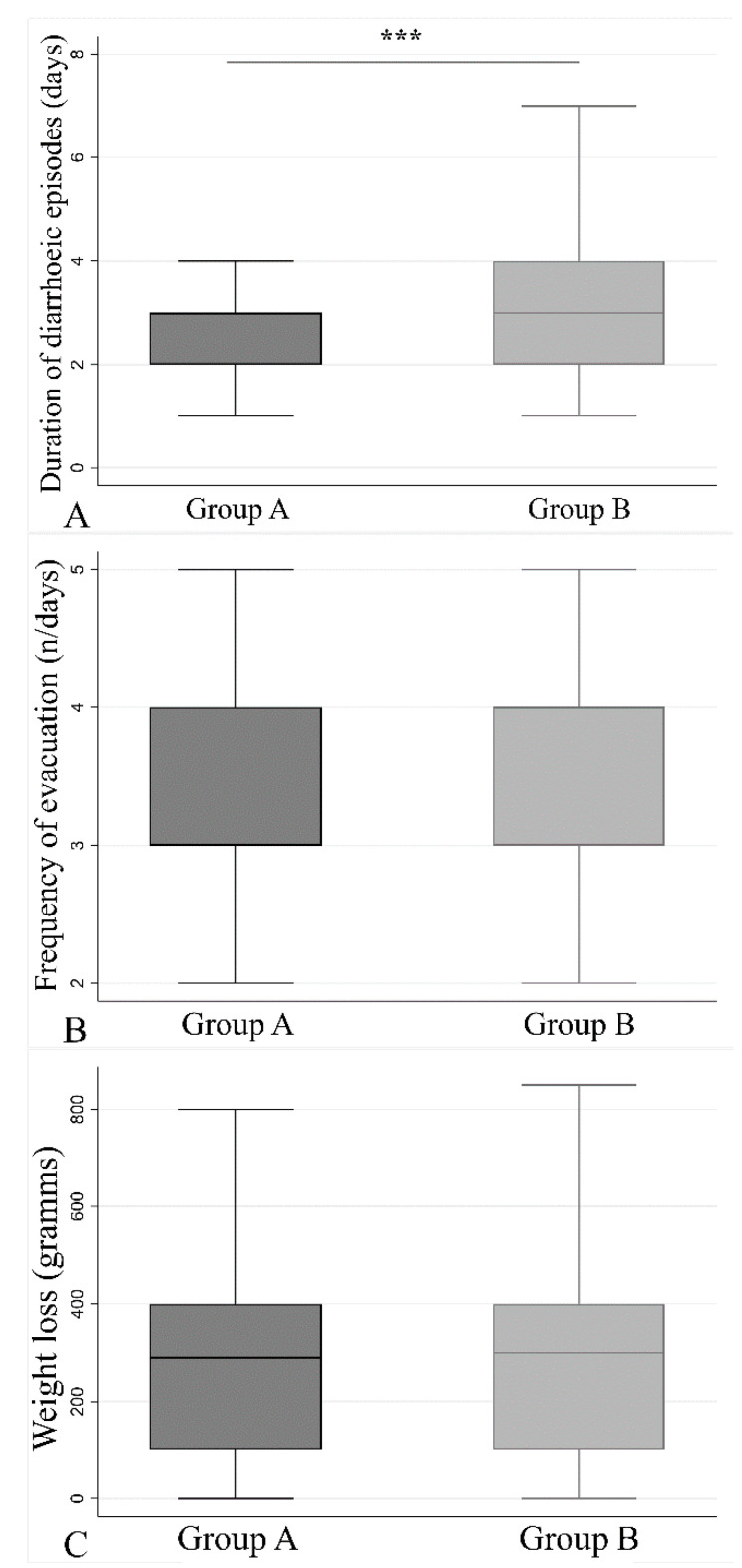
Physiological parameters recorded for each group. Student’s t test was used to compare the two groups among the following parameters: number of days of antibiotic-associated diarrhea (AAD) is less in group A with respect to group B (**A**) (*p* < 0.001). The frequency of evacuation (**B**) and the weight loss (**C**) do not differ between the two groups, *p* = 0.856 and *p* = 0.512, respectively. (*** *p* value: <0.001).

**Figure 3 jcm-09-03080-f003:**
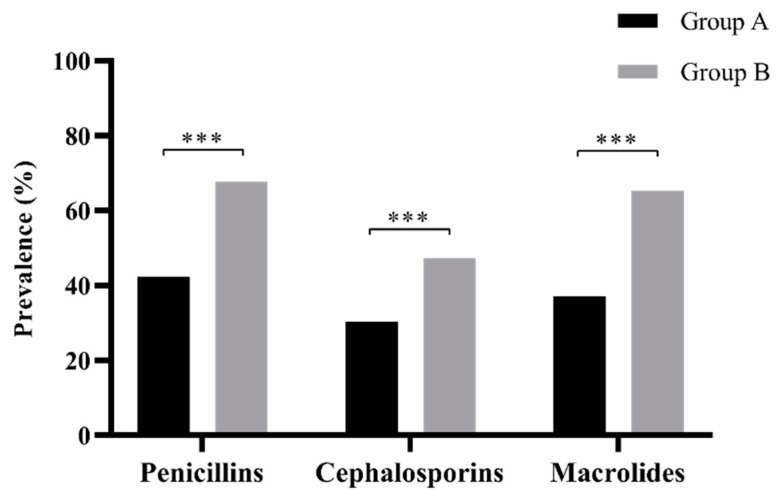
Prevalence of AAD in group A and group B, with respect to the type of antibiotic administered. A statistical difference among the three antibiotic categories resulted between the two groups (*p* < 0.0001). For more details, refer to Appendix A (*** *p* value: <0.001).

**Figure 4 jcm-09-03080-f004:**
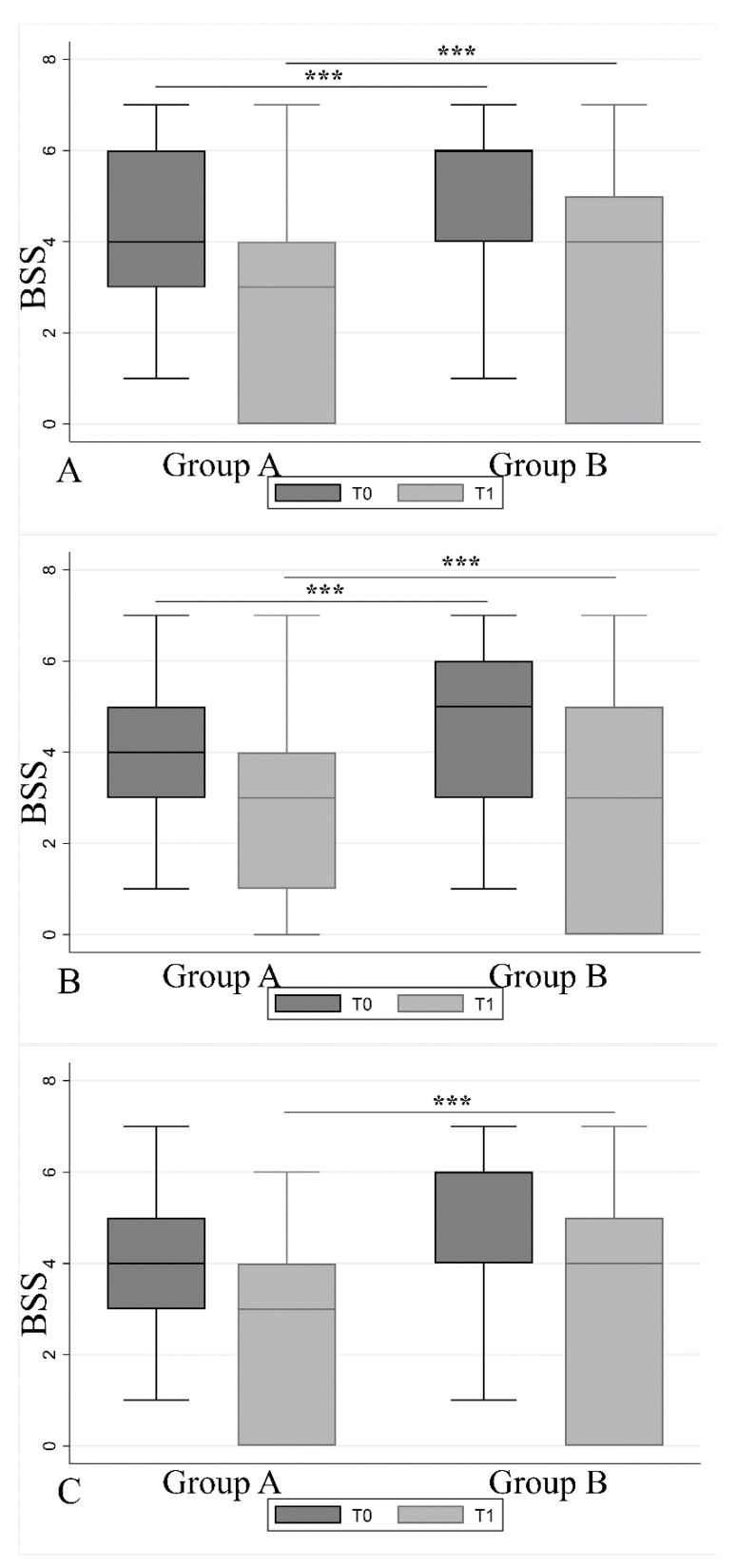
Bristol Stool Scale comparison among the three antibiotic categories over T0 and T1. One-way ANOVA (between group A and B) followed by multiple comparison (Tukey’s multiple comparisons test) found strong statistical difference among (**A**) penicillins, (**B**) cephalosporins, and (**C**) macrolides at the end of the antibiotic treatment (T0) and 30 days after (T1). (*p* < 0.0001.) (*** *p* value: <0.001.) For more detail, refer to Appendix A.

**Figure 5 jcm-09-03080-f005:**
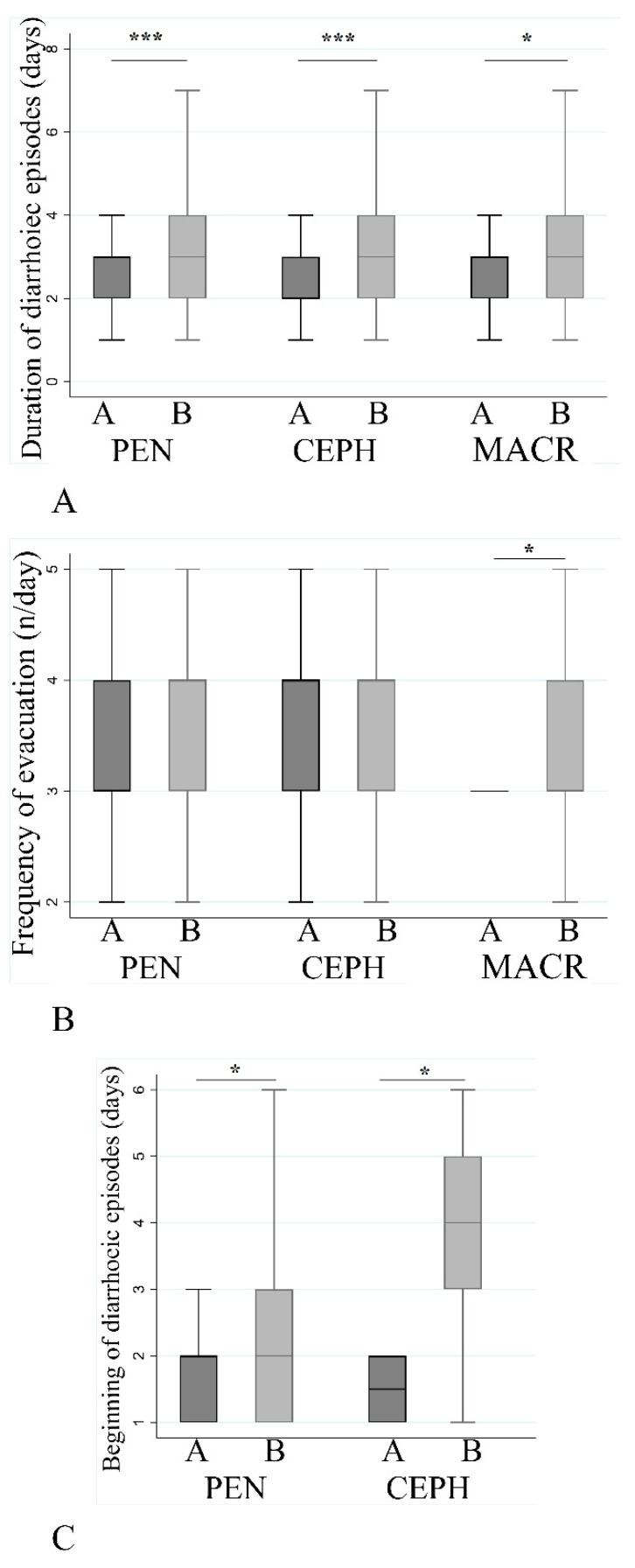
Effect of antibiotic on different diarrhea associated parameters. Comparison between two groups was made using a Student’s *t* test. Duration of diarrhea (number of days) (**A**), number of stools per day (**B**), and beginning of diarrhea (**C**) with respect to the three antibiotic categories. (* 0.05 < *p*-value < 0.01, *** *p* value: <0.001.) Abbreviations: PEN = penicillins, CEPH = cephalosporins, MACR = macrolides.

**Table 1 jcm-09-03080-t001:** Overall distribution of the infections among the population.

Infections	Prevalence; N (%)
Upper Respiratory Tract Infections (URTI)	6786 (68.1%)
Lower Respiratory Tract Infections (LRTI)	1610 (16.2%)
Urinary Tract Infections (UTI)	620 (6.2%)
Otitis (Ot)	567 (5.7%)
Gastroenteritis (Ge)	143 (1.4%)
Other	234 (2.4%)

**Table 2 jcm-09-03080-t002:** Distribution of clinical conditions between group A and group B.

Infections	Group A; N (%)	Group B; N (%)	*p* Value
URTI	3390 (67.2%)	3396 (69.1%)	0.0357
LRTI	788 (15.6%)	822 (16.7%)	>0.05
UTI	325 (6.4%)	295 (6%)	>0.05
OT	314 (6.2%)	253 (5.1%)	0.0238
Ge	100 (2%)	43 (0.9%)	0.0002
Other	131 (2.6%)	103 (2.1%)	>0.05
Total	5048	4912	

Comparison of differences in proportion of patients with different clinical conditions between treated and control groups, by Chi-square test. For each abbreviation, see Table 1.

**Table 3 jcm-09-03080-t003:** Comparison of stool consistency (Bristol Stool Scale) in AAD patients at the end of antibiotic treatment (T0) and 30 days after (T1).

	Time	Group A	Group B	*p* Value
Bristol Stool Scale	T0	4.6 (± 4.4)	5.4 (± 4.7)	<0.001
	T1	3.7 (± 1.3)	4.4 (± 1.4)	<0.001
*p* value		<0.001	<0.001	
Difference T1-T0		−0.9 (SD 4.4)	−1.2 (SD 4.8)	0.007

Group A shows better improvement in the score compared to group B. The comparison was carried out using a Student’s *t*-test; results are shown as mean ± standard deviation.

**Table 4 jcm-09-03080-t004:** Comparison of stool consistency using the Bristol Stool Scale stratification at T0 and T1.

T0				T1			
BSS	Group A	Group B	*p* Value	BSS	Group A	Group B	*p* Value
1	141 (3.5%)	84 (2.3%)	0.001	1	185 (3.7%)	127 (2.6%)	0.002
2	306 (7.7%)	235 (6.3%)	0.023	2	382 (7.6%)	309 (6.3%)	0.012
3	885 (22.2%)	539 (14.5%)	<0.001	3	757 (15.0%)	457 (9.3%)	<0.001
4	1229 (30.8%)	633 (17.1%)	<0.001	4	1427 (28.3%)	1013 (20.6%)	<0.001
5	532 (13.3%)	480 (12.9%)	0.626	5	633 (12.5%)	919 (18.7%)	<0.001
6	623 (15.6%)	1170 (31.5%)	<0.001	6	127 (2.5%)	481 (9.8%)	<0.001
7	279 (7.0%)	568 (15.3%)	<0.001	7	26 (0.5%)	101 (2.1%)	<0.001

Data were grouped according to the BSS score assigned to each group. The statistical comparison between group A and group B was carried out using a Chi-squared test; in bold are the statistical differences.

**Table 5 jcm-09-03080-t005:** Prevalence of prescribed antibiotics among group A and group B.

Antibiotics	N (%)	Group A, N (%)	Group B, N (%)	*p* Value
Penicillins	4774 (47.9%)	2423 (48.0%)	2351 (47.9%)	>0.05
Cephalosporins	3804 (38.2%)	1942 (38.5%)	1862 (37.9%)	>0.05
Macrolides	884 (8.9%)	419 (8.3%)	465 (9.5%)	0.0407
Other	35 (0.4%)	21 (0.4%)	14 (0.3%)	>0.05
Missing	463 (4.7%)	243 (4.8%)	220 (4.5%)	>0.05

Statistical differences were evaluated for each of the categories using a Chi-squared test. The prevalence of macrolides in group B was statistically significant (*p* = 0.0407) compared to group A.

**Table 6 jcm-09-03080-t006:** BSS score in AAD patients.

	Time	Group A	Group B	*p* Value
Penicillins				
Bristol Stool Scale	T0	4.8 (±4.3)	5.7 (±5.0)	<0.001
	T1	3.8 (±1.1)	4.4 (±1.3)	<0.001
Difference T1–T0		−1.0 (±4.4)	−1.4 (±5.0)	0.096
Cephalosporins				
Bristol Stool Scale	T0	4.3 (±4.0)	5.0 (±4.3)	<0.001
	T1	3.6 (±1.2)	3.9 (±1.4)	<0.001
Difference T1–T0		−0.7 (±4.0)	−1.1 (±4.4)	0.009
Macrolides				
Bristol Stool Scale	T0	5.1 (±6.5)	5.6 (±5.4)	0.251
	T1	3.6 (±1.2)	4.4 (±1.2)	<0.001
Difference T1–T0		−1.5 (±6.4)	−1.6 (±5.4)	0.820

Stratification between the three classes of antibiotic. A Student’s *t*-test was used to compare the two groups, results are expressed as mean and standard deviation.

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
