# Peer review of "Effect of Limosilactobacillus reuteri LRE02–Lacticaseibacillus rhamnosus LR04 Combination on Antibiotic-Associated Diarrhea in a Pediatric Population: A National Survey"

_jcm, 2020, doi:10.3390/jcm9103080_

Round 1
Reviewer 1 Report
General Comments:
The manuscript of Lorenzo et al describes the effect of combination probiotics on antibiotic-associated diarrhea. The authors have shown that treatment of pediatric patients with AAD with both Limosilactibacillus reuteri and Lacticaseibacillus rhamnosus as probiotics for 30 days can reduce the incidence of AAD and decrease the number of days of diarrhea.
The topic addressed is interesting and deserves a constructive discussion. I think however that there are a few improvements that should be made before publication.
Specific comments.
- Why did Group A develop diarrhea earlier than Group B? Please add a little more to the discussion session on why Group A has an earlier onset of diarrhea than Group B.
- Do you have an analysis of the gut microbiota before and after administration of probiotics in some of the subjects in this study?
- Please add a detailed description of the Bristol Stool Scale.
- I cannot find the Ethical Review Board approval number in the text of this article. The author needs to add the IRB approval number.
- Line 284: ~ in group B (2.8±1.3 288 days vs 3.2±1.4 days), depending on~. What does the number 288 days mean?
Author Response
Dear reviewer,
thank you for the comments and suggestions to our study. Here a point by point response to your answers
Answer one: Why did Group A develop diarrhea earlier than Group B? Please add a little more to the discussion session on why Group A has an earlier onset of diarrhea than Group B.
Response one: We did not find any other explanation in the literature to support other hypotheses. It should be interesting to mimic gastrointestinal epithelium and study this specific interaction in vitro but this scenario goes beyond the specific aims of this study.
Answer two: Do you have an analysis of the gut microbiota before and after administration of probiotics in some of the subjects in this study?
Response two: It should be useful to understand the changes in microbiota composition before and during the probiotic administration to monitor how effectively probiotics exert their biological functions, however this is not the aim of this work.
Answer three: Please add a detailed description of the Bristol Stool Scale.
Response three: we added this description
Answer four: I cannot find the Ethical Review Board approval number in the text of this article. The author needs to add the IRB approval number.
Response four: informed consent was obtained from the patient’s parents according to the principles outlined in the Declaration of Helsinki. Moreover, the Institutional Review Board (made by different pediatricians in the national territory) gave us a positive approval
Answer five: Line 284: ~ in group B (2.8±1.3 288 days vs 3.2±1.4 days), depending on~. What does the number 288 days mean?
Response five: we deleted this typographical error

Reviewer 2 Report
This is an interesting open label randomized study assessing the prevalence of AAD after a specific probiotic treatment. The study is well written and easy to read but suffers from the absence of use of placebo and the patients and investigators blinding.
However, before a possible acceptance, the authors should address the following points:
- Throughout the paper there are several typos including underlined words.
INTRODUCTION
- The authors should better explain the reason why they choose LRE02 and LR04 instead of other probiotics and the advantages of this mixture over other probiotic strains.
- If there only a company producing the probiotic used, please cite the commercial name.
DESIGN
- The author should report whether informed consent was obtained and if the study was approved by ethical committee
- IT would be interesting to have data of diarrhea and other GI symptoms after 1 month of follow up. I expect that under probiotic treatment the patient will remain asymptomatic or with very few symptoms. Instead, it is interesting to assess whether diarrhea come back after the treatment.
- Are there inclusion and exclusion criteria for patients included in the present study?
- Have been excluded patients with other GI diseases?
- Have been diagnosed or investigated other underlying GI conditions?
- A stratification for weight loss on weight age-percentile is needed.
- A separate analysis for infants and children and adolescent is needed.
RESULTS
- A table reporting general, demographics and clinical characteristics of patients included in the survey is needed.
- Weight loss should be stratified per infants and children and adolescent. As you know infants with diarrhea experience different and more heavy weight loss compared to adolescent.
- BSS of patients of group A is significantly different from that of group B. This may have biased the result of the study. Moreover, in both cases (group A and B) at T1 there is an improvement in BSS. Moreover, table 3 and 4 are a sort of repetition of figure 1. Please keep only the most informative.
- I do not understand the reason for comparing the prevalence of antibiotics used between the two groups. What is the clinical meaning?
DISCUSSION
Line 236: dysbiosis in the first year of life has been associated even with IBD and celiac disease.
CONCLUSION
Line 298 : you have not evaluated adverse effects. Please remove it from conclusions or explain with results.
Author Response
Dear reviewer,
thank you for the comments and suggestions to our study. Here a point by point response to your answers
Answer one: Throughout the paper there are several typos including underlined words.
Response one: all these typos were removed
Answer two: The authors should better explain the reason why they choose LRE02 and LR04 instead of other probiotics and the advantages of this mixture over other probiotic strains.
Response two: a short explanation was added in the discussion. The main advantage of these probiotics relies in the microencapsulation process that increases the bioavailability and survival of the encapsulated strains compared to the classical freeze-dried formulations available. For the specific reasons for probiotics choice, in vitro preliminary studies (data not shown in the main text) demonstrated that the formulation of LRE02 and LR04 showed better results in terms of co-culturing and survival of both the species.
Answer three: If there only a company producing the probiotic used, please cite the commercial name.
Response three: this detail was added.
Answer four: The author should report whether informed consent was obtained and if the study was approved by ethical committee
Response four: informed consent was obtained from the patient’s parents according to the principles outlined in the Declaration of Helsinki. Moreover, the Institutional Review Board (made by different pediatricians in the national territory) gave us a positive approval.
Answer five: IT would be interesting to have data of diarrhea and other GI symptoms after 1 month of follow up. I expect that under probiotic treatment the patient will remain asymptomatic or with very few symptoms. Instead, it is interesting to assess whether diarrhea come back after the treatment.
Response five: These data were not recorded.
Answer six: Are there inclusion and exclusion criteria for patients included in the present study?
Response six: the inclusion/exclusion criteria were added in Material and methods
Answer seven: Have been excluded patients with other GI diseases?
Response seven: We excluded patients with acute and/or chronic diarrhea before enrollment, diagnosis of chronic gastrointestinal diseases or diagnosis of other chronic diseases.
Answer eight: Have been diagnosed or investigated other underlying GI conditions?
Response eight: An exclusion criteria is referred to the presence of diagnosed chronic GI diseases.
Answer nine: A stratification for weight loss on weight age-percentile is needed.
Response nine: We think that adding another stratification could interfere with the aims of this study. Moreover, ages of patients are not homogeneously distributed and this could negatively affect the statistical analysis.
Answer ten: A separate analysis for infants and children and adolescent is needed.
Response ten: As stated in the previous answer, we did not perform this analysis because data are not homogeneously distributed to draw out any statistical result. No adolescents were included in this study
Answer eleven: A table reporting general, demographics and clinical characteristics of patients included in the survey is needed.
Response eleven: It is impossible to resume all the patient’s information in a single table. Also, a supplemental table could be not suitable.
Answer twelve: Weight loss should be stratified per infants and children and adolescent. As you know infants with diarrhea experience different and more heavy weight loss compared to adolescent
Response twelve: No adolescents were included in this study
Answer thirteen: BSS of patients of group A is significantly different from that of group B. This may have biased the result of the study. Moreover, in both cases (group A and B) at T1 there is an improvement in BSS. Moreover, table 3 and 4 are a sort of repetition of figure 1. Please keep only the most informative.
Response thirteen: We added to table 3 and 4 to support our results based on the request of another reviewer
Answer fourteen: I do not understand the reason for comparing the prevalence of antibiotics used between the two groups. What is the clinical meaning?
Response fourteen: The incidence of diarrhea can be very different in relation to different antibiotics. Some antibiotics have an important effect on intestinal motility, such as amoxicillin, which can, more easily, cause diarrhea. For this reason, a separate analysis was made.
Answer fifteen: dysbiosis in the first year of life has been associated even with IBD and celiac disease
Response fifteen: thank you for this advice.
Answer sixteen: you have not evaluated adverse effects. Please remove it from conclusions or explain with results.
Response sixteen: we have removed this sentence.

Round 2
Reviewer 1 Report
The authors have substantially revised this manuscript and have answered all of my suggestions/concerns.
Author Response
Thank you for all the suggestions you gave us.
Reviewer 2 Report
I read with interest the review made by the authors which improved the quality of the manuscript.
However, as previously stated a key-point for a correct interpretation of the results of the paper is the stratification according to the age of patients included, since the authors stated that have been enrolled patients from 1 month to 18 years old.
Moreover, demographics characteristics of patients included according to age categories (example 0-5 years, 5-10 years and >10 years) is of absolute importance for scientific correctness. We don't know anything about these patients except the diagnosis, the age range and the BSS.
Author Response
Dear reviewer,
We have modified the main text and added a supplemental table (Supplemental Table 1), which includes the stratification for the weight loss according to your request. Unfortunately, we have no other demographics characteristics to add.
This manuscript is a resubmission of an earlier submission. The following is a list of the peer review reports and author responses from that submission.
Round 1
Reviewer 1 Report
In this study by Lorenzo et al, the suitability of using a combination of Lactobacillus reuteri LRE02-DSM 23878 and Lactobacillus rhamnosus LR04-DSM 16605) is tested against a large cohort of patients for their effect on the prevalence of AAD in the presence of antibiotics. Moreover, of those patients demonstrating AAD, several factors including stool consistency (BSS), AAD duration, frequency of evacuation, beginning of diarrhoeic episodes and weight loss were investigated. This also included an evaluation of these factors with respect to the type of antibiotic given to the patients.
This study has merit and the data is better presented, however there are still a number of issues to address.
- Title should only have the new species name. The former name can be listed in the Introduction. With that, the authors have written it incorrectly. They need to list the new name and then in brackets (only the first time) put the former name. ie. Limosilactobacillus reuteri (formerly name Lactobacillus Reuteri LRE02).
- The authors must not write the dosage as 'five drops per day' (this means nothing) and instead put the actual cfu delivered per day. This has to be changed throughout the text in the paper.
- The authors must include their main findings and conclusions at the bottom of their introduction (as clearly stated in the Instructions to authors and by me in my initial review).
- Also in the Introduction:
- add a reference for "AAD needs more appropriate monitoring and specific medications.
- Incorrect referencing for Goldenberg, Guo and McFarland. All should have et al and then the year listed. ie. Goldenberg et al (2015). Also italicise the et al.
- In the Materials and Methods, the authors state that the pediatricians are located 'at the national territory. Do they mean 'in the National Territory (Phillipines)?
- Also in the Statistical Analysis part of their Materials and Methods, the authors need to include that they used a One-way Anova. They also need to state what multiple comparison was used for that test.
- In the Results section:
- The authors state that for Table 2, the statitical significant difference regards only proportion of patients with URTI and OT, 67.2 vs 69.1 an d6.2 vs 5.1 respectively. I still cannot see the differences here as most of the numbers for the percentages in brackets are missing. Also, the authors need to state that the percentages and therefore the p value reflects each infection over the number of total infections. This is not clear here. I thought they were comparing Gp A and GpB for each individual infection. ie. N = 3390 vs 3396 for URTI.
- The text on all axes of graphs in Fig 2, Fig 4 and Fig 5 are too small.
- Also in Fig 2A, the median line is missing from the Gp A box plot and for both Gp A and B in Fig 2B.
- The authors must state what multiple comparison test was used to generate the data in Fig 4.
- Again, no median is present for T0 GpB in Fig 4A and 4C. Also, the data looks identical for GpB in Fig 4A and 4C. How is this possible?
- Fig 5 cannot be right. All box plots between GpA and GpB in Fig 5A and between groups in Fig 5B (except Gp A macrolides, where the box plot is missing altogether) are identical. How can that be? And if so, why are the macrolides showing * when the other two categories are showing *** in the Fig 5A graph where the box plots are identical?
- The boxes for the box plot of Gp A (macrolides) in Fig 5B and 5C are missing? As are the median lines in various box plots throughout Fig 5. Also, the authors need to say why the data for macrolides in Fig 5C is na? Do they mean NS (not statistically significant)?
- The Fig 5 legend also requires attention. 3 p values are shown in brackets but 0.01 is missing the asterisks etc.
- The authors need to be consistent throughout their paper. Either write T0 and T1 or Time 0 and Time 1. This also applies to Group A and Group B (sometimes it's written as group A and group B, while other times it's written as Group A and Group B).
- The authors should align all the units in the p value column in Table 5.
- In the Discussion:
- The authors must include a reference for 'Dr Metchnikoff opened the 'era' of probiotics...'
- The authors must state which strains were used in the statement '...the recommended daily serve should contain 109 live cells of at least one of the strains'.
- The authors must include a reference for their statement '....an imbalance to the microbiota can potentially result in the occurrence of gastrointestinal or extraintestinal disease'.
- The authors need to reference the following statement '..which when compared to other probiotics, is one of the most appropriate strains for preventing AAD due to its modest NNT.
- Again, 'Lactobacillus rhamnosus GG (LGG) (formerly Lacticaseibacillus rhamnosus)' is written around the wrong way. ie. Lacticaseibacillus rhamnosus was formerly known as Lactobacillus rhamnosus GG (LGG).
- The authors need to explain further what Microencapsulation increases resistance to in the Discussion.
- The authors state that 'The most prescribed antibiotics were penicillins, followed by cephalosporins and macrolides, according to the recent AIFA National report'. The authors should include what these antibiotics were used to treat or at least mention that these antibiotics are used to treat a range of infections.
- In the Discussion where the authors have written 'Group A than in group B (2.8±1.3 days vs 3.2±1.4 days)', which antibiotic category are they referring to. Also, these numbers (2.8 vs 3.2) are not evident from the figure).
- In References: Some of the references need to be fixed as they still contain words written in Italian.
- There is alot of phrasing throughout your paper that needs attention. The authors should engage a person that is well versed in the English language to help them with this.
Author Response
Please see the attached file.
Regards

Reviewer 2 Report
Despite the change to ANOVA with box plots - I don't see that the results look different enough to have p-values almost exclusively of P<0.001. I also am concerned that the authors found their own previous methods "were not suitable to understand statistical comparison" and I am not sure now that a one-way ANOVA is any better. Further, there are no edits to the statistical methods section to justify the use of an ANOVA or that any changes in statistical methodology occurred.
Author Response
Please see the attached file.
Regards.

Round 2
Reviewer 1 Report
In Title
Add ‘Effect of a Limosilactobacillus……on antibiotic-associated diarrhea in a pediatric population’ in your title.
In Abstract
State to who the questionnaires were delivered to by each patient/parent during the visits after antibiotics and probiotic treatment in your abstract.
Abstract still contains the words ‘five drops per day for 30 days’. Change this.
Change ‘independently by the type of antibiotic. Moreover, the use of probiotics allowed a normal stool consistency in a shorter time, evaluated with BSS’. These 2 sentences need rephrasing.
- independent of the type of antibiotic used. Moreover, the use of probiotics resulted in a normal stool consistency in a shorter time period, as evaluated by the BSS.
In Introduction:
“Mostly, it is a mild condition requiring no treatment, and resolving on their own’. Should read on its own.
‘concern being also the burden and costs not well documented by national surveillance studies’. Again, this statement in the introduction needs to be rephrased. ie. …..concern being that the burden and costs are also not well documented by national surveillance studies.
Change ‘Lacticaseibacillus rhamnosus (formerly Lactobacillus rhamnosus) or Saccharomyces boulardii at 5- 40 billion colony forming units (CFU)/day could be appropriate to improve NNT’ to
‘Lacticaseibacillus rhamnosus (formerly Lactobacillus rhamnosus) or Saccharomyces boulardii at 5- 40 billion colony forming units (CFU)/day appeared to improve the NNT’.
Change to something like this:
‘In particular, dosage appeared to account for the substantial reduction of the NTT. The authors also reported the subgroup effect based on high dose probiotics (≥ 5 billion CFUs per day), which revealed an NNT of 6 (12). Overall, these results confirm the importance of strain specificity and the dose-related effect of probiotics as shown by McFarland et al (2018)’.
Change to something like this:
‘CFU of both microorganisms daily for 30 days) was shown to be associated with a decrease in AAD in children, and had no known adverse effects. Besides these results, a reduction in AAD prevalence, number of days with diarrhea and a significant improvement in stool consistency was demonstrated.
In Materials and Methods:
Change to:
‘and delivered to a Medical Doctor (MD) who collected the data in an electronic online repository’.
In Results:
Table 1 still has the value of the % missing for URTI in Gp B!
Change to:
‘Stool consistency results are reported in Figure 1. At T0, in group A the mean value of the BSS score was 4.5±1.5 (median 4, IQR 3-5), while in group B it was 4.9±1.6 (median 5, IQR 4-6) (p value<0.0001)’.
Bottom of Fig 1:
Change ‘The comparison between two groups at T0 and T1 was made by t-Student test…’ to
“The comparison between two groups at T0 and T1 was made using a Students t-test…
At the bottom of Table 3 change to: Group A shows better improvement in the score compared to group B. The comparison was done using a Student’s t-test; results are shown as the mean ± standard deviation.
Change to ‘In group A, diarrhea occurred in 1.7 days following treatment, while in group B this occurred after 2.1 days….
Change ‘Besides, the number of days with diarrhea in group A (2.8±1.3 days) was significantly lower than those reported in group B (3.2±1.4 days)’ to ‘Additionally, the number of days with diarrhea in group A….
Change Fig legend for Table 4 from
‘Data were grouped according to the BSS score assigned to each group. The statistical comparison between group A and group B was done with Chi-squared test, in bold are the statistical differences’. to
‘Data were grouped according to the BSS score assigned to each group. The statistical comparison between group A and group B was done using a Chi-squared test; in bold are the statistical differences’.
Also say ‘using a Chi-squared test’ for the Fig legend underneath Table 5.
Change Figure 3 title to: Prevalence of AAD in group A and group B, with respect to the type of antibiotic administered.
For Figure 4 place the letters ‘A’ ‘B’ and ‘C’ at the top left-hand corner of each graph.
For Figure 5, all tables you provide in your response should go into supplementary information.
I also think that the median lines in your box plots for Fig 5A in particular, need to be thicker to show the differences between the 3 antibiotic categories.
For Fig 5C Macrolides, as you only have n=1 this part of the graph must be removed. If this is the same for Fig 5B Macrolides, then this part of the graph must also be removed. You cannot draw any conclusions from just 1 patient.
Change Fig 5 legend to: Comparison between two groups was made using a Student’s t-test. Duration of diarrhea (number of days) (A), number of stools per day (B), and beginning of diarrhea (C) with respect to the three antibiotic categories.
And for: ‘The stool consistency in AAD-positive patients (N=4897) at T0 and T1 shows statistical differences between the two groups with respect to the three antibiotic categories (Table 5). Comparison between the two groups was made by using a Student’s t-test’.
At the bottom of Table 5 change to: ‘Stratification between the three classes of antibiotic. A Student’s t test was used to compare the two groups; results are expressed as the mean ± standard deviation’.
In Discussion:
Change to this: (NOTE: That I have removed some of your words as well in some parts for the phrasing to be correct).
‘The World Health Organization defines probiotics as "live microorganisms which when administered in adequate amounts confer a health benefit on the host" (Food and Agriculture Organization and World Health Organization, 2001). Dr. Metchnikoff opened the "era" of probiotics because he was the first to propose that ingesting certain bacteria could help replace harmful microbes in the body (16). Based on various studies, the minimum quantity sufficient to obtain temporary colonization of the intestines by a microbial strain is 109 live cells per day (17) (Need at least 3 References here). Therefore, in the case of miscellanea, the recommended daily consumption should contain 109 live cells of at least one of the strains including Saccharomyces cerevisiae or lactic acid fermenters such as Streptococcus thermophilus or Lactobacillus bulgaricus (17,18)’.
‘Moreover, the use of antibiotics in the first years of life interferes with the microbiota development and results in dysbiosis (25).
It is also noteworthy that dysbiosis related to antibiotic use in the first years of life is a risk factor for obesity (27), functional gastrointestinal disorders (28) and impaired neurocognitive outcome (29).
Change to: A Probiotics' mechanism of action is exerted through the modulation of the content of gut microbiota, maintenance of the integrity of the gut barrier, prevention of bacterial translocation and the modulation of the local immune response by the gut-associated immune system. In particular, Lacticaseibacillus rhamnosus is a probiotic strain, which when compared to other probiotics, is one of the most appropriate for preventing AAD due to its modest NNT (30).
Fix:
In our randomized intervention study, we evaluated the efficacy of a new double strain probiotic formulation which included Limosilactobacillus reuteri (formerly Lactobacillus reuteri LRE02) 2×108 CFU and Lacticaseibacillus rhamnosus (formerly Lactobacillus rhamnosus LR04) 1×109 CFU on the prevalence of AAD. These probiotic strains are free of plasmids that carry antibiotic resistance, are microencapsulated, and have a shelf life (i.e. live and viable bacteria) that is guaranteed for 24 months.
These next 2 statements need references:
‘Microencapsulation increases the resistance of probiotic microorganisms during
the gastro-duodenal transit, thus ensuring their titer and biological activity (REFS). At the same time, it protects cells from degradation phenomena due to external factors (humidity, acidity, osmotic pressure, oxygen, and light) (REFS)’.
Change to:
As stated in two different studies by Del Piano et al (2011 and 2012), the microencapsulation of probiotics offers numerous advantages compared to the freeze-dried technique; the most important of which is the high gut colonization as a result of the increased number of viable cells that transit the intestine and, as a consequence, the reduction of the probiotic concentration delivered in these lipidic microcapsules due to a strong gastro-resistance of the cells (33,34).
You need to further explain what you mean by the statement I have highlighted in yellow.
Change to:
The type of antibiotics and the reason for their administration do not appear to affect the outcome of the probiotic on AAD, as previously reported for one of two probiotics recommended by ESPGHAN for AAD prevention (1,37).
Change to:
In AAD patients, the mean BSS score at T0 was not different between the two groups, regardless of the antibiotic categories used; however, at T1, the BSS score was significantly better in group A than in group B. Surprisingly, in AAD patients, diarrhea started earlier in group A than in group B, regardless of the type of antibiotic used. This effect could be related to the influence of gastrointestinal commensals on motility as previously observed in vivo and ex vivo in mice and rat models, in which the administration of L. reuteri and different Lactobacillus species where shown to moderate jejunal motility within minutes (38–40).
Fix this reference in your discussion: ‘Guarino (2008)’ as for the other references.
Change to:
‘which studied an in vitro model of human colonic cells exposed to L. rhamnosus GG and found a significant shortening of smooth muscle cells that had an impact on motility (41). Evidence is accumulating on the hypothesis that certain probiotic strains have acute actions in vivo and ex vivo on the host's autonomic reflexes and can act differently on the small compared to the large intestine (39).
Change to and add this sentence to the previous paragraph. You have too many short paragraphs:
These effects can occur within minutes, suggesting that the inter-kingdom signaling responsible for them do not rely on colonization, alteration in the microbiome composition, or any other longer-term adjustments (42).
Explain what you mean by longer term adjustments (highlighted in yellow)?
Change to:
However, the number of days with diarrhea was significantly lower in group A than in group B (2.8±1.3 288 days vs 3.2±1.4 days), depending on the antibiotic used, as macrolides showed the worst duration. We did not find any difference in the number of stools/day and weight loss in AAD patients in either of the two groups. This is in line with the other studies that have shown that probiotics are associated with a reduction in the mean duration of AAD of 18 h, without having an effect on the numbers of stools per day (23, 10).
Change:
This study has demonstrated that using our probiotic combination from the beginning of antibiotic therapy is also helpful in reducing the mean duration of diarrhea (calculated in days) as well as in obtaining a normal stool consistency in a shorter time.
In conclusions:
Remove the word ‘daily’ or ‘per day’ from ‘daily per day’ for 30 days. They both mean the same thing.
Change to:
‘is associated with lower rates of AAD in children (aged one month to 18 years) without any increase in adverse effects. A reduction in AAD prevalence, the reduced number of days with diarrhea and a better stool consistency was demonstrated regardless of the type of antibiotic used or the reason for their prescription. However, a double-blinded clinical trial is required to further confirm these results.
Reviewer 2 Report
This is an RCT of (control) antibiotics alone vs (treatment) antibiotics + double-probiotics in AAD. While the concept is meritorious, there is no description of the baseline status of the control vs treatment group to confirm that the randomization worked. Instead the data begins at T0, which I think is 5 days after the treatment or control condition has been administered. At 5 days (T0) there are many significant differences between the treatment and control group - including significant differences in the outcomes of interest. These differences could easily be because the randomization did not work but there is no way to tell because the only baseline data that is presented are the clinical conditions at baseline - which actually show significant differences between control and treatment groups.
-There needs to be a description of all outcomes of interests, basic demographics, etc. at baseline for the control and treatment groups to verify if the randomization worked (note: Baseline is usually described as T0 in most studies)
-There needs to be a timeline of when antibiotics and probiotics were given vs. the nomenclature of timepoints - were all antibiotics given for the same length of time? Were the probiotics given for the same length of time? Were the probiotics taken at the same time as the antibiotics or advised to be taken at a different time of day?